# Collaborative Unmanned Vehicles for Inspection, Maintenance, and Repairs of Offshore Wind Turbines

**Mohd Hisham Nordin** [1,*], **Sanjay Sharma** [1], **Asiya Khan** [1], **Mario Gianni** [2], **Sulakshan Rajendran** [1] **and Robert Sutton** [1]

1   Autonomous Marine Systems Research Group, School of Engineering, Computing and Mathematics, University of Plymouth, Plymouth PL4 8AA, UK; sanjay.sharma@plymouth.ac.uk (S.S.); asiya.khan@plymouth.ac.uk (A.K.); sulakshan.rajendran@plymouth.ac.uk (S.R.); r.sutton@plymouth.ac.uk (R.S.)
2   Centre for Robotics and Neural Systems, School of Engineering, Computing and Mathematics, University of Plymouth, Plymouth PL4 8AA, UK; mario.gianni@plymouth.ac.uk
*   Correspondence: mohdhisham.nordin@plymouth.ac.uk

**Abstract:** Operations and maintenance of Offshore Wind Turbines (OWTs) are challenging, with manual operators constantly exposed to hazardous environments. Due to the high task complexity associated with the OWT, the transition to unmanned solutions remains stagnant. Efforts toward unmanned operations have been observed using Unmanned Aerial Vehicles (UAVs) and Unmanned Underwater Vehicles (UUVs) but are limited mostly to visual inspections only. Collaboration strategies between unmanned vehicles have introduced several opportunities that would enable unmanned operations for the OWT maintenance and repair activities. There have been many papers and reviews on collaborative UVs. However, most of the past papers reviewed collaborative UVs for surveillance purposes, search and rescue missions, and agricultural activities. This review aims to present the current capabilities of Unmanned Vehicles (UVs) used in OWT for Inspection, Maintenance, and Repair (IMR) operations. Strategies to implement collaborative UVs for complex tasks and their associated challenges are discussed together with the strategies to solve localization and navigation issues, prolong operation time, and establish effective communication within the OWT IMR operations. This paper also briefly discusses the potential failure modes for collaborative approaches and possible redundancy strategies to manage them. The collaborative strategies discussed herein will be of use to researchers and technology providers in identifying significant gaps that have hindered the implementation of full unmanned systems which have significant impacts towards the net zero strategy.

**Keywords:** collaborative; unmanned vehicles; UAV; USV; UUV; offshore wind turbine; unmanned operations; localization

## 1. Introduction

The Operation and Maintenance (O&M) of a Wind Turbine (WT) are very important. Within the O&M activities, the maintenance part plays a crucial role to ensure the longevity of a WT as well as to lower the cost of its daily operation. Proper maintenance activities such as regular inspections and repairs will reduce WT downtime, thus reducing losses in energy output [1]. However, maintaining a WT requires extra effort due to the complexity of tasks performed at a high-rise tower and the hazardous environments it is operating in. These levels of complexity increase tenfold concerning the Offshore Wind Turbine (OWT) due to its location. O&M of an OWT is more challenging than the normal WT due to the sea waves' action and high wind which could damage the OWT structure [2]. Even more, human operators are also exposed to more hazardous environments at sea during the transit and accessing times of the asset and during the Inspection, Maintenance, and Repair (IMR) operations. Due to the increase in Health and Safety (H&S) issues for

human operators as well as to minimize the cost associated with IMR activities, unmanned operations are viewed as promising solutions which also lead towards the net zero strategy to reduce significant carbon footprint by 2050 [3].

Although Unmanned Aerial Vehicles (UAVs), Unmanned Underwater Vehicles (UUVs), and Unmanned Surface Vehicles (USVs) have already been deployed for unmanned operations of the OWT [4,5], they are mostly used for visual inspection operations such as for Wind Turbine Blade (WTB) and subsea surveys. Most major industries have begun to deploy unmanned techniques in their IMR activities; however, this is not the case for the OWT industries, probably due to the major challenges they face, such as being located far from land, within the treacherous sea environments, as well as having limited capabilities in current unmanned technologies. Unmanned systems can be classified into fully autonomous, semi-autonomous, teleoperation, and remote control [6]. The commonly used mode for current unmanned operations at the OWT is remote control, where a UAV is used for visual WTB inspection and Remotely Operated Vehicles (ROV) for underwater surveys. However, full autonomy for inspections using UAVs and UUVs is rapidly being deployed today. Nonetheless, the deployment of these autonomous vehicles also faces greater challenges and issues, under Beyond Visual Line of Sight (BVLOS), especially with energy and communication issues [7]. Hence, these limitations further hinder the potential for conducting unmanned IMR for OWT.

Several efforts have been initiated to advance the potential of implementing unmanned IMR for OWT. Among the solutions are to install a localized power charging station near the OWT farms using fixed stations [8–10] or using a docking vessel either crewed [11] or uncrewed [12,13]. The latter solution is seen as moving forward due to many benefits it brings, including reducing carbon footprint, increasing operator safety by remaining onshore, and increasing efficiency with continuous operations [14]. Furthermore, the use of uncrewed vessels also functions as a communication hub that connects the UAVs or UUVs to the onshore Ground Control Station (GCS) [15]. A recent technological breakthrough in terms of real-time communication has shown a positive trend in deploying uncrewed vessels for offshore energy activities. The presence of an uncrewed vessel or a USV near an OWT farm has enabled a new kind of unmanned network that has the potential to be utilized in a collaborative manner. With sound collaboration strategies between multiple UVs, it has the potential to solve many issues surrounding the IMR operations of the OWT.

Collaborative UVs have been reviewed in other fields such as surveillance [16–18] and search and rescue missions [19–21], as well as in agricultural activities [22,23]. However, to the best of our knowledge, no review article has been published within the OWT field. Nonetheless, several concept papers and simulation work related to collaborative UVs for OWT have been presented [24–26]. In contrast to other previous review work related to the implementation of collaborative UVs, this paper:

- Reviews the works that have the potential to be implemented within the OWT IMR operations.
- This paper also highlights many strategies that can be implemented as a major enabler toward reaching this goal, including strategies for UVs to perform complex tasks, achieve better localization and navigation, prolong operation time, and establish effective communication.
- Reviews are conducted mostly on related works ranging from 2018 up to 2022.
- Summarizes the main findings related to the potential solutions and improvements a collaborative UVs approach may bring to the unmanned IMR OWT operations.
- Briefly discusses the potential failure modes and redundancies for UVs in the IMR OWT operations and further highlights the importance of exploring redundancies within this area to ensure the reliability, availability, and safety of such collaborative approaches.

This paper is structured as follows: Section 2 reviews the state-of-the-art capabilities of current UVs already being deployed in other sectors, focusing on their potential to be implemented within the OWT sector; Section 3 presents the unmanned collaborative

strategies that can be implemented by utilizing multiple types of UVs to perform IMR operations for the OWT; and Section 4 briefly discusses the potential failure modes and redundancies for UVs in the IMR OWT operations. Finally, a conclusion and future directions related to research areas in collaborative UVs are also presented.

## 2. The State-of-the-Art of UVs Performing in OWT IMR Operations

The types of UV currently available for unmanned OWT inspection are either remotely piloted [27,28] or autonomous [29] and can either be a UAV [30], USV [13], or UUV [31]. Unmanned inspections carried out by UAVs are mostly performed using visual-based approaches [28–30,32]. However, UAVs which can perform close-contact Non-Destructive Testing (NDT) inspections are also available [33], although they are very limited. While the UAVs perform inspections for OWT structures above sea level, the UUVs on the other hand are currently being used to perform unmanned inspections for underwater OWT structures [34–36]. Meanwhile, unmanned inspections using USVs are also being carried out for the purpose of surveying underwater seabed for OWT farms [13], as well as monitoring the scouring of pile foundations [5]. However, these multiple types of UVs are currently known to operate independently, hence limiting the potential or capability to perform unmanned OWT maintenance or repair works.

Although it is easier for a UV to perform a visual inspection of an OWT, the lack of extra hands or tools, as compared with manual operation by human operators, means that it proves to be a very difficult task for a single UV to perform maintenance or repair work. For example, a UAV with limited capability to carry heavy equipment or to carry a set of different tools will not be able to perform maintenance or repair work. A USV on the other hand, would not be able to reach a very high-rise OWT tower to repair the blades, or reach an undersea OWT structure or undersea cables. Meanwhile, a UUV might be able to perform maintenance and repair works on subsea cables or OWT undersea structures but would still require a nearby vessel or station to charge its batteries, for tethering purposes or to relay inspection and maintenance data.

To enable extra capabilities to perform maintenance and repair, a UAV built with dual 5-axis robotic arms has been developed [37]. This platform has the capability of carrying up to 10 kg of payloads, thus enabling the ability to carry and to manipulate tools for IMR purposes. This UAV can also be capable of changing tools during flight. However, the maximum flight time is only 30 min, which might be sufficient for most maintenance tasks but is insufficient for OWT deployment. The remotely operated UAV will also have to depend on a nearby supporting vessel, as the flight time might be insufficient for offshore operations. Moreover, a human operator is needed onboard the vessel to charge the batteries.

Alternatively, drone swarms for OWT operations are currently being designed and developed [38,39]. However, the solutions remain off-limit for maintenance and repair purposes as these drones are only capable of performing visual inspections, though at a faster rate or at a wider scale. Moreover, a swarm system needs a very complex algorithm to perform maintenance or repair in a collaborative manner. Nevertheless, several efforts utilizing multiple drones in other fields have been observed, with the potential to be implemented for WT maintenance and repairs such as for carrying heavy payloads [40–42] or assembling structures [43].

Moving forward, significant efforts which are highly viewed as the most viable solution are currently being developed, which relate to the integration of multiple types of UVs that can be implemented to collaboratively perform unmanned OWT IMR (e.g., UAV and USV [44–53], USV and UUV [54–57], UAV and UUV [58,59], or a combination of all UVs [60–63]). In addition, the use of crawler robots has also been proposed in this collaborative system to increase the capability and flexibility of WT maintenance and repairs [24].

Table 1 briefly summarizes the works that have been put forth related to collaborative unmanned vehicles. Basically, it highlights the relevant contributions for each work which have the close potential to be implemented within the OWT IMR operations.

**Table 1.** Summary of UVs collaboration types, purposes, and methods.

| Types | Ref. | Purposes | Methods | | Application |
|---|---|---|---|---|---|
| **UAV-USV** | [44] | Recovery | - | Using a fiducial marker located on a USV for accurate UAV landing. | General |
| | [45] | Recovery | - | Using a three-stage fiducial marker on a USV to improve landing stage detection by UAVs. | General |
| | [46] | Communication platform | - | Using a distributed dynamic network topology for fulfilling effective communication for establishing required formation between UAVs and USVs based on an ad hoc network. | General |
| | [47] | Path planning, navigation, communication platform | -<br>- | Using aerial mapping and real-time aerial visual assistance provided by UAV for USV path planning and safe cruising.<br>Using datalink telemetry as a communication relay when a USV is in a GNSS-denied area. | Monitoring of rivers and dams |
| | [48] | Recovery, transportation | -<br>-<br>- | Using adjustable buoys and a unique carrierdeck for safe landing and transporting of a UAV on a USV.<br>Using 3D path planning by generating a sequence of guide points for a UAV towards the USV deck.<br>Using integrated ultrasonic sensors on a USV to ensure UAV landing positioning accuracy. | General |
| | [49] | Recovery | -<br>-<br>- | Using an infrared receiver on a UAV to detect infrared beacons on a USV.<br>Using a two-phase UAV precise landing method to land on a USV.<br>Using USV control and path-following algorithms to guide UAV. | General |
| | [50] | Launch and recovery | -<br>- | Using a robotic recovery system for a water-landing UAV onto a USV.<br>Using a solar-powered automated USV platform. | General |
| | [51] | Communication platform | - | Using location-based beam steering algorithm to establish high-speed communication links between a GCS, a USV, and a UAV. | General |
| | [52] | Power supply | - | Using a wireless charging pad on a USV platform to charge UAV's batteries. | Monitoring of water pollution |
| | [53] | Power supply | - | Using power umbilical tethers from a USV to supply power which results in longer UAV flight time. | General |
| **USV-UUV** | [54] | Localization | - | Using an Extended Kalman Filter (EKF) augmented by ultra-short baseline (USBL) range and visual-data-based localization from a USV to enhance UUV localization. | General |
| | [55] | Launch and recovery | -<br>-<br>- | Using a stern platform to recover a UUV onto a USV.<br>Using a sonde transmitter for communication from a USV to a UUV Central Unit.<br>Using optical communication from a UUV to the USV to send back sonar data and underwater positions. | Surveying of ports and critical infrastructure |
| | [56] | Recovery | - | Using visual lighting and acoustic guidance strategies for the Docking and Line Capture Line Recovery (LCLR) system to recover a UUV onto a USV. | General |

**Table 1.** *Cont.*

| Types | Ref. | Purposes | Methods | | Application |
|---|---|---|---|---|---|
| | [57] | Localization, power supply, communication hub | - | Using a waypoint tracking algorithm on a USV to provide relative heading and coordination to UUV. | Underwater survey |
| | | | - | Using an underwater cable to supply power from a USV to a UUV and to transfer a large amount of data from a UUV to a USV. | |
| **UAV-UUV** | [58] | Navigation | - | Using the position on the aerial view acquired from a UAV to manually guide a UUV towards a target. | Survey and waste management |
| | [59] | Communication platform | - | Using an acoustic device attached to a UAV, data can be transferred from a UUV directly to a floating UAV. | Underwater survey |
| | [60] | Mission planning, path planning, localization, navigation | - | Using visual imagery from a UAV and a UUV, a USV acts as data and an intelligent hub to autonomously plan and distribute survey missions. | Surveying of floating targets |
| | | | - | Using a USV GNSS positioning system to guide the UUV through underwater communication. | |
| | | | - | Using underwater visuals to guide USV from colliding with underwater obstacles. | |
| **UAV-USV-UUV** | [61] | Path planning | - | Using a cooperative search algorithm based on random simulation experiments and asynchronous planning strategies, an underwater target can be detected by a UAV, USV, or UUV. | Underwater, search-and-track mission |
| | | | - | Using the information of the detected target position, a UUV will track the underwater target. | |
| | [62] | Communication platform | - | Using an underwater acoustic channel characterization to enable effective communication between UAV, USV, and UUV. | General |
| | [63] | Power supply, communication platform | - | Using tethered connections from a USV to UAVs and UUVs to prolong operation time. | Inspection and survey missions on offshore infrastructures |
| | | | - | Using Wi-Fi and UHF antennas on a USV, data from UAV and USV can be communicated to the onshore ground control station. | |
| **USV-UAV-CR** | [24] | Transportation, power supply | - | Using a USV with battery charging capability to transport UAVs and CRs to an offshore wind farm. | IMR for offshore wind farms. |
| | | | - | Using a UAV or UAVs to carry CRs to an inspection site. | |

From Table 1, it can be summarized that collaborative UVs can be used to solve many challenging issues surrounding unmanned operations such as communication, launch and recovery, localization and navigation, power supply, and transportation. Most of the works focus on solving unmanned operations that can be applied to many areas without focusing on a specific application. However, several works have demonstrated using UVs in a collaborative manner to solve unmanned operations in areas such as surveying, search and track, monitoring, and inspection. Up to the current date, no research work has demonstrated successful implementation of collaborative UVs within the OWT sector, rather focusing on future views and directions [24].

By deploying collaborative UVs, it can immensely simplify and solve many issues faced by a single UV or by manual operators in performing OWT maintenance and repairs. The major enablers or opportunities that might arise have the ability to perform complex IMR tasks as well as gaining a more accurate and reliable localization and navigation system. However, to realize these opportunities, the challenges that come with them need to be addressed as well, which are discussed in the next section.

### 3. Implementing Collaborative UVs for the OWT IMR Operations

Collaboration between similar or different types of unmanned vehicles, aerial, surface, or underwater, to perform IMR tasks for OWT can be established through various strategies. Currently, the level of feasibility needed to implement collaborative UVs for OWT IMR operations is only between two different UVs. Even though the state-of-the-art multi-drone platform has a single control interface to support a UUV and a UAV [63], however, the collaboration is either between a UAV and a USV or between a UUV and a USV. Therefore, to successfully implement collaborative UVs for the OWT IMR operations, one must establish sound strategies between any two UVs in the first instance. The important strategies that can be devised are in performing complex tasks, solving localization and navigation issues, prolonging operation time, and establishing effective communication.

*3.1. Strategies to Perform Complex Tasks*

UAVs on their own can be utilized to inspect high-rise structures, including wind turbines. Currently, they are being used for non-contact visual inspection and NDT ultrasonic inspection [33]. However, a great amount of effort still needs to be carried out for a single drone to perform inspections with high task complexity, such as having to carry heavy payloads (sensors and actuators) or performing inspections in inaccessible areas (narrow spaces or difficult angles). For maintenance or repair operations, the task complexity is even higher as it might involve performing a series of operations with different tools. Hence, a single UAV performing repair or maintenance work on a high-rise wind turbine structure is seen as far-fetched due to the current capabilities of a single UAV.

The level of complexity to perform maintenance tasks ranges from very low complexity up to very high complexity. Performing visual inspection can be considered a very low complexity task while, in contrast, gearbox replacement activities are at a very high complexity level. Unmanned technology for undertaking very high complexity tasks such as replacing wind turbine blades or gearbox does not exist yet and is currently deemed unsuitable. Apart from that, there are still many opportunities for unmanned operations to be deployed for OWT, which mostly depend on the level of complexity. In most cases, a single UV is insufficient to perform unmanned maintenance tasks, thus requiring collaborative strategies to be deployed.

One of the sources of complexity arises when a task requires using multiple tools for completion. For instance, repairing surface damages on a wind turbine blade usually involves the processes of cleaning, sanding, filling and putting, coating, and painting [64]. The strategies that can be implemented to perform this type of task are fulfilled by using multiple homogenous UAVs equipped with a different set of tools [65] or by using a single UAV with the capability of carrying and changing tools for different tasks [66], as illustrated in Figure 1a,b. Both strategies will depend on a USV as a docking and power charging station; however, a tool-changing UAV will rely more on a USV as a tool changing station. For a tool-changing UAV, it may also be a UAV that is equipped with a manipulator [66,67], an arm [43,68,69], or multiple arms [37,70] to perform more dexterous manipulation during maintenance or repair works. Table 2 summarizes the different features and potential collaboration or adaptation methods that can be achieved with these types of tool-changing UAVs.

**Table 2.** Potential collaboration/adaptation methods between a USV and a tool-changing UAV.

| End Effector | Ref. | Special Features | Potential Collaboration/Adaptation Methods |
|---|---|---|---|
| Manipulator | [66] | An elastic gripper with a grip and release mechanism. | An automated tools magazine can be placed on a USV platform where a UAV will fly over it and select a desirable tool to exchange by grasping or releasing. |
| | [67] | A 2-Degree-of-Freedom servomotors-based gripper with 360 degrees of rotation. | With a rotating gripper mechanism, a UAV will have greater flexibility to pick up tools on a USV which is placed without fixed orientation. |

**Table 2.** *Cont.*

| End Effector | Ref. | Special Features | Potential Collaboration/Adaptation Methods |
|---|---|---|---|
| Single-arm | [43] | For crawler deployment and retrieval. | Using a 6-Degree-of-Freedom robotic arm integrated with a UAV capable of retrieving a crawler robot on a specialized platform onboard a USV. |
| | [68] | Replaceable robotic arm. | Several robotic arms can be placed on a USV where a UAV will have the flexibility to choose a suitable single robotic arm (different degrees of freedom) for different applications. |
| | [69] | Equipped with a gimbal and a dynamic gravity compensation mechanism. | This UAV will have greater control flexibility while acquiring a tool onboard a swaying USV. The mechanism allows the motion of a robotic arm to be decoupled from a drone, ensuring flight stability while the arm is being operated. |
| Multiple arms | [37] | Dual robotic arms for carrying up to 10 kg of payload. | This UAV has the ability to fetch and carry bigger or longer tools which are transported on a USV for the purpose of IMR operations. |
| | [70] | Equipped with three robotic arms with ultrasonic sensors for landing on uneven surfaces. | This UAV has the ability to carry and transport more objects to IMR sites. It also has the potential to perform preprocessing tasks such as preassembling objects while still onboard a USV. |

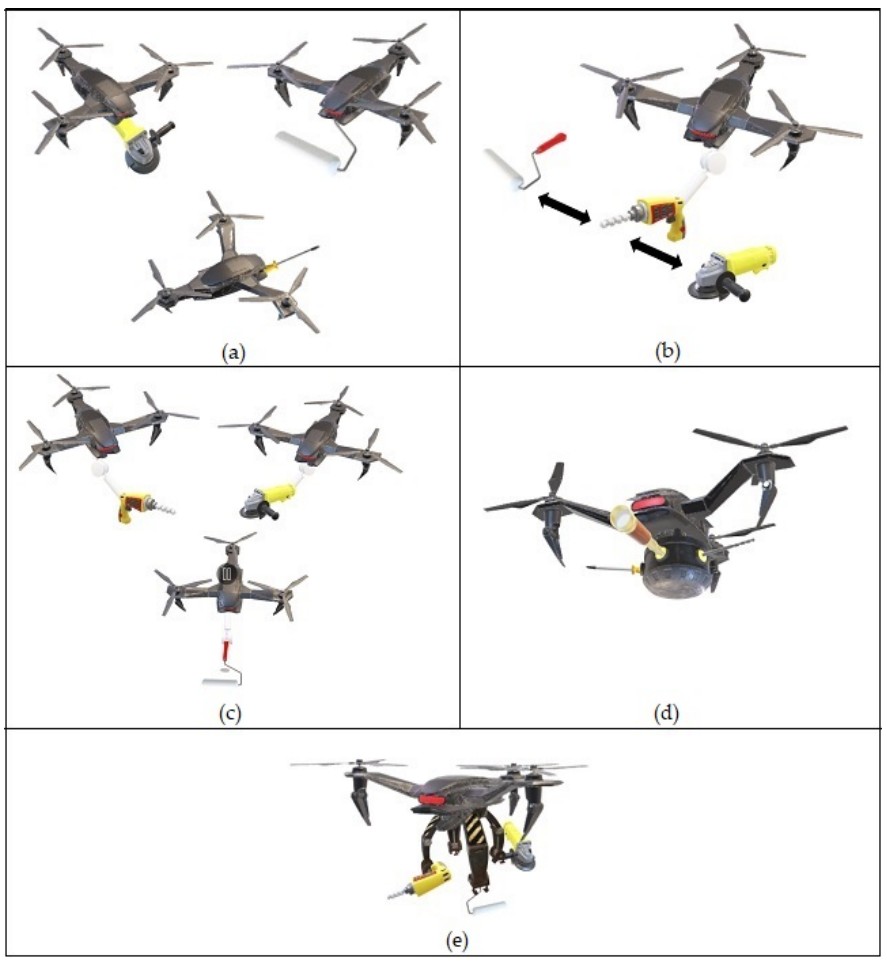

**Figure 1.** Different types of UAVs to perform complex tasks: (**a**) multiple homogenous UAVs with different fixed tool, (**b**) single UAV with multiple tools changing, (**c**) multiple UAVs with multiple tools changing, (**d**) single UAV with a magazine of tools, and (**e**) single UAV with multiple robotic arms holding different tools.

To increase the flexibility in using different tools for limited assets, the multiple tool-changing UAVs (Figure 1c) can also be implemented as substitutes for the multiple fixed-tool UAVs. There is also a possibility of introducing a UAV which carries a magazine with multiple tools (Figure 1d) or a UAV with multiple arms carrying different tools (Figure 1e), providing that the payload capacity is manageable. The illustrations in Figure 1 show different types of UAVs that can be deployed in this strategy, while Table 2 summarizes the potential collaborations or adaptation methods between a USV and a tool-changing UAV.

Using multiple UAVs which hover close to each other during a maintenance operation, a task can be accomplished relatively fast by quickly taking turns to perform different roles such as drilling [71], cleaning [72], coating [73], or painting [74,75]. A careful sequence of operations can be planned for multiple UAVs or for multi-robot [76] performing such operations by modelling roles and task allocation within a cooperative heterogeneous system. Based on many-sorted first-order logic, assigning roles and tasks for heterogeneous multi-robot operations can be achieved using temporal relations. The temporal relations enable expressing a wide class of collaborative tasks among the robots. To reason on the multiple choices faced during role and task assignment, tensor decomposition has been proposed.

However, due to hovering instability caused by high wind, these UAVs need to attach themselves tightly to the blade during operation. A UAV with the capability of attaching itself firmly to a WTB or any structure either horizontally or vertically is of great advantage. To land and stick firmly on an irregular surface as on a WTB, a mechanism that uses two swivel arms with four vacuum suction cups for a UAV can be implemented [77]. The two swivel arms can be adjusted accordingly, using a gear system that also acts as a counterbalance to ensure greater stability during operations. Alternatively, a crawler robot (CR) that can attach firmly to a WTB can also be deployed by a UAV [24], using a gripping mechanism for deployment and retrieval. To stick to the WTB, this crawler robot also uses vacuum suction cups as a gripping mechanism attached to each of its six legs.

For underwater maintenance and repair operations, similar approaches to UAV-USV collaboration can be devised for implementing a collaborative USV-UUV. A UUV can be equipped with a manipulator, a robotic arm, as well as multiple robotic arms [78]. Regarding a tool-changing UUV [79], it already exists and has been commercialized for underwater IMR activities. To facilitate the tool-changing operation, this UUV depends on an underwater facility that stores a set of tools. This underwater facility is purposely built to enable continuous operations without the presence of a tools-carrying vessel, making it greener. However, problems might arise in certain situations such as operating in a vast underwater area and difficult human access to tools for maintenance or repair. In a vast underwater area such as under OWT farms, the distance a UUV needs to travel back and forth for tool-changing will not be practical. Using a collaborative USV-UUV approach, a green USV can be carefully designed to be equipped with a submersible tool-changing facility. If any of the tools need to be replaced, repaired, or maintained through human intervention, the USV can simply return to the onshore base.

There are many collaborative strategies between UVs that can be planned and implemented for unmanned OWT IMR operations. In most cases, the involvement of a USV in any collaboration will assist a UAV or UUV in terms of functioning as a docking and power charging station [50,80], providing extra computing power and data storage capacity [60,61], as a communication hub between all other UVs and GCS [25,51], as well as providing a reliable localization and navigation system [60], as will be further discussed in this section. Table 3 presents a summary of the benefits and challenges associated with different collaboration strategies. The challenges that may arise are also presented in the table.

### 3.2. Collaborative Localization and Navigation Strategies

The accuracy, precision, and reliability of the UV localization and navigation system are important within the IMR activities of an OWT. Without localization accuracy within

centimetre or millimetre range, a UV will still be capable of navigating without suffering collision with other objects. In addition, the implementation of multiple onboard sensors such as LIDAR, infrared obstacle detection sensor, Time-of-Flight sensors, ultrasonic sensor, or stereo vision [81] may prevent collision effectively. However, for a UV to work within the IMR activities, localization accuracy within the centimetre or millimetre range is important to accurately identify and report inspection data that need further attention. Working near a WT that is susceptible to losing signal or suffering measurement error related to GNSS, a coordinate for inspection data provided by a UAV might not be reliable. Even worse, for a UUV working underwater in a GNSS-denied environment coupled with low visibility, obtaining reliable and accurate IMR coordinates is even more challenging. As a result, operators or UVs might face difficulties in locating the exact position for future revisiting due to the measurement inconsistency.

**Table 3.** Collaborative strategies using different types of UVs for OWT M&R operations.

| Collaborative UVs | Strategies | Benefits | Challenges |
|---|---|---|---|
| USV-UAV | A USV carrying a fixed-tool UAV | • UAV may perform well in a specialized task. | • Requires different UAVs for different applications.<br>• Requires several trips onshore to change UAV with a different tool. |
| | A USV carrying a tool-changing UAV | • UAV may perform multi-task operations.<br>• Different sets of tools can be stored on a USV. | • Requires tools cartridge system onboard the USV.<br>• Requires several trips to USV for tool changing. |
| | A USV carrying multiple arms UAV | • More flexible to perform high-dexterity tasks.<br>• May carry and work with several tools per trip.<br>• Different sets of tools can be stored on a USV. | • Difficult for a fully autonomous system.<br>• Requires uninterrupted communication with the GC station through the USV.<br>• Requires tools cartridge system onboard the USV. |
| USV-UAVsUSV-UAVs | A large USV carrying multiple fixed-tool UAVs | • Can perform multiple operations at different spots simultaneously.<br>• Can perform multiple tasks in sequence using different sets of tools across several UAVs.<br>• Can be used to lift heavy tools using multiple UAVs. | • Requires high coordination and scheduling algorithm.<br>• Difficult to handle a task requiring manipulations. |
| | A large USV carrying multiple tool-changing UAVs | • Can be used to lift heavy tools using multiple UAVs.<br>• Tools can be shared between UAVs | • Extensive back and forth trips to change tools.<br>• Requires high coordination and scheduling algorithm. |
| | A large USV carrying multiple-arm UAVs | • Can be used to lift heavy tools.<br>• Collaborative tasks between UAVs can be established. | • Requires a very good uninterrupted communication system between human operators. |
| | A large USV carrying heterogeneous UAVs | • Can perform multiple operations simultaneously with greater flexibility. | • Difficult to coordinate multiple types of UAVs working collaboratively. |

**Table 3.** *Cont.*

| Collaborative UVs | Strategies | Benefits | Challenges |
|---|---|---|---|
| USV-UUV | A USV carrying a tool-changing UUV | • Can perform multi-task operations.<br>• Different sets of tools can be stored on a USV.<br>• Option for using tethered UUV. | • Requires tools cartridge system beneath the USV.<br>• Requires several trips to USV for tool changing.<br>• Requires a UUV recovery mechanism. |
| | A USV carrying UVMS | • More flexible to perform high-dexterity tasks.<br>• Can carry and work with several tools per trip.<br>• Different sets of tools can be stored on a USV.<br>• Option for using tethered UVMS. | • Difficult for a fully autonomous system.<br>• Requires uninterrupted communication with the GC station through the USV.<br>• Requires a UVMS recovery mechanism. |
| USV-UUVs | A large USV carrying multiple tool-changing UUVs | • Tools can be shared between UUVs. | • Requires a recovery and storing mechanism for the UUVs.<br>• Tethered UUVs might become tangled with one another. |
| | A large USV carrying multiple UVMSs | • Collaborative tasks between UVMSs can be established. | • Requires a very good uninterrupted communication system between human operators.<br>• Requires a recovery and storing mechanism for the UVMSs.<br>• Tethered UVMSs might become tangled with one another. |
| | A large USV carrying heterogeneous UUVs | • Collaborative tasks between UUV and UVMS can be established. | • Requires universal recovery and storing mechanisms for the UUVs.<br>• Tethered UUVs/UVMSs might become tangled with one another. |
| Multiple USVs-UAVs/UUVs | Multiple USVs carrying UAVs or UUVs | • Handling mechanism for a single UAV/UUV is manageable. | • Requires complex coordination between USVs and other UVs.<br>• Requires centralized control strategy from one of the USVs. |
| UAV-Crawler | UAV carrying a robotic crawler | • Robotic crawler is more stable to perform IMR tasks. | • Requires a complex coordination/localization system to retrieve or deploy a robotic crawler on an OWT. |

For a UAV operating at a low height near a wind turbine structure, a GNSS is prone to fail due to multipath errors, where the receiver might read a reflecting signal instead of a direct signal [82]. To solve this problem, the visual localization and navigation method [83–87] is seen as a viable solution which is summarized in Table 4.

Another method that can be used to achieve reliable coordinates for IMR data is by anchoring the UAV localization system to radio transmitters or specific points near an OWT structure. Having accuracy within centimetres resolution, ultra-wide band technology (UWB) is suitable to be applied for UAV localization in a GNSS-denied environment [88]. Using four UWB modules as fixed anchors to a UAV carrying a UWB transceiver, the coordinate of the UAV can be obtained [89,90]. However, for OWT operation, the use of fixed anchors is not suitable, as the sea environment is always in motion. Nevertheless,

anchors with auto-calibrated capability [91] have enabled mobile platforms to be used as anchors' placement for UAV localization.

**Table 4.** Visual localization methods for UAV working near a wind turbine.

| Ref. | Localization Methods | Descriptions |
|------|---------------------|--------------|
| [83] | Simultaneous model fitting and pose estimation. | Positional accuracy can be achieved through precise image measurements which are based on an optimization algorithm. |
| [84] | Robust visual-inertial odometry with point and line features. | A faster line detector and purified strategies enable the detection of wind turbine blades by extracting sufficient features. Fusion of IMU and point-line features exhibit higher accuracy. |
| [85] | Visual SLAM with a synthetic depth map. | Using a monocular SLAM system, a visual-based localization system to detect wind turbine blades is improved using a synthetic depth map. The depth map is based on a Line Segment Detector. |
| [86] | Autonomous visual navigation algorithm using image processing methods and blade feature detection method. | Using Wavelets, Wiener filtering for a deblurring system, image enhancement using Retinex, and Hough transform for wind turbine edge detection. Blade feature detection is performed by comparing the extracted features (circle and linear features) with the features from a blade library. |
| [87] | Visual SLAM. | Focussing on localization within the internal wind turbine blade, localization and navigation can be achieved using the Visual SLAM method. To build a map for localization and navigation, V-SLAM uses distinguished features such as edges, corners, or colours. |

A possible collaborative solution that can be implemented using mobile UWB anchors is by utilizing multiple USVs as platforms for UAV localization. In this configuration, at least four USVs can be deployed to float around an OWT as shown in Figure 2. Three-dimensional positioning of a UAV from the UWB modules can be achieved using the multilateration method [92,93] with a GNSS receiver installed on each of the USV. As for the positioning of the USVs around the OWT, they can be placed as far as the UWB signals can reach to minimize the effect of operating within a GNSS-denied environment.

Currently, the commercialized UWB positioning devices for an outdoor environment have accuracy smaller than 5 cm within the range of 50 m and 75 m [94]. Their signal may reach up to 200 m but with lower accuracy [95], thus they are deemed less practical to be implemented for the time being since the average height of OWT is 100 m [87]. Nevertheless, disruptive innovation in the UWB range measurement using microwave radar has been presented where a very long-range measurement up to 6 km with accuracy below 10 cm has been achieved [96]. It is therefore expected that a mid-range high-accuracy UWB positioning system with a measurement range beyond 100 m will be available in the near future.

For UUV localization, the UWB localization method is not suitable since it is based on a radio frequency which cannot be used for underwater applications. Instead, centimetre-level accuracy for the UUV coordinate can be achieved using the long baseline (LBL) method [97]. However, due to the nature of the LBL which needs to be fixed to the ocean bed, the detection range becomes limited for the UUV. To solve this issue, the inverted LBL method can adopt the multilateration technique by implementing four USVs and an AUV [97] as depicted in Figure 3. Within this configuration, all the USVs are equipped with GNSS receivers and ultra-short baseline (USBL) modems for transmitting and receiving acoustic signals. Meanwhile, the AUV is also fitted with a USBL modem. A practical approach for a full OWT IMR package could easily be realized by integrating both the UWB and the inverted LBL localization methods on the USVs.

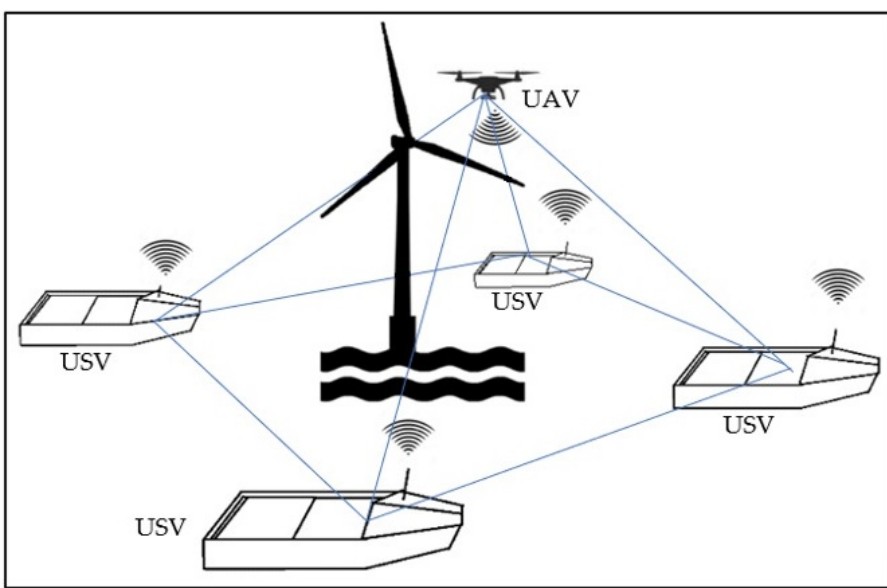

**Figure 2.** Configuration for multiple USVs-UAV using mobile UWB anchors.

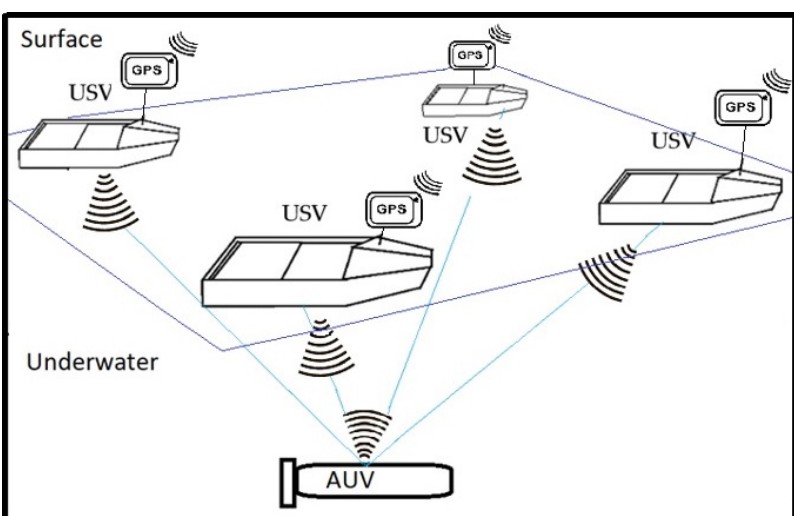

**Figure 3.** Inverted LBL approach.

By integrating the GNSS with Real-Time Kinematic positioning (RTK) [98], a centimetre accuracy level can be achieved for the UV localization system. Basically, in the GNSS-RTK system, at least two receivers are used where one of them is stationary (base station) and fixed on a precisely known location [99]. For OWT farms located more than 60 km from a base station, the RTK approach cannot be used as it already reached the maximum operational distance. Alternatively, using Post-Processed Kinematic (PPK) [100], the maximum operating distance can be up to 100 km; however, the PPK system is not suitable for control operations as it is not a real-time system. Another alternative is by using the Network RTK (NRTK) which can provide centimetre-level accuracy below the 100 km range from a reference station [101]. Unfortunately, the NRTK method depends on available inland network reference stations, which do not exist at sea.

Since more OWT farms are moving further from the land, the prospect of using the GNSS-RTK method is diminishing. However, Precise Point Positioning (PPP) technique for correcting GNSS position in real time can be used to provide centimetre-level accuracy without depending on reference stations [102]. Several articles integrating the PPP technique with GNSS for UAV have been reported [103–105]. Nevertheless, the effectiveness and the accuracy of operating PPP techniques for using GNSS positioning close to tall

structures or wind turbines have not been explored yet. It has been discovered that the PPP is vulnerable to poor GNSS observations, cycle slips, and data outages [105]. Therefore, it can be assumed that the PPP method is also susceptible to inaccurate measurement when working in an area prone to GNSS signal loss, such as working closely to a tall structure or a wind turbine.

Since the IMR operations are more concerned with the localization of the OWTs instead of the UVs, a wind turbine coordinate as an absolute coordinate system should be used. Using the wind turbine coordinate, dependencies on the GNSS can be reduced to only necessary navigation while approaching the OWTs. However, for a swaying OWT [106], as in the case of a floating OWT, its platform is greatly influenced by the wind and waves. A flight path generated from the initial absolute coordinate might not be accurate, as the final position on the OWT might already sway. Thus, a positioning system for a UV needs to be constantly updated with respect to the absolute coordinate of the OWT.

For a swaying WT, a virtual point to represent an origin for the absolute coordinate (0,0) can be initially marked by a UAV. Once the origin is set, the UAV can either be path-planned (autonomous system) or remotely navigated to the point of interest. Due to the swaying motion of the floating OWT, a correctional plan for the UAV localization system needs to be executed based on the current position and orientation of the floating OWT. A correctional approach for UAVs based on ground truth navigation can be devised utilizing UWB anchors that are placed at the OWT base. Equipped with IMUs, this UWB device can provide the floating OWT with its current orientation and motion displacement. The information from this device can then be transmitted to the UAV for real-time relative localization corrections.

However, the localization system that depends on the GNSS or UWB positioning does not provide robust navigation since the wind turbine causes magnetic disturbance [89]. Alternatively, a localization system that is not susceptible to any radio interference is seen as the more robust solution. Methods such as visual-based [83,85,107] and simultaneous localization and mapping (SLAM) [108–110] are gaining interest as the more reliable localization approach when working near an OWT. In a visual-based localization, irrespective of the swaying motion of the OWT, a UAV can detect the origin, which is virtually set earlier. Therefore, it can then easily correct the localization of any point of interest on the OWT with respect to the origin. Since a wind turbine consists of moving components as well as stationary structures, the selection of the origin's location is crucial.

To localize any point on the WTB, the centre hub of a wind turbine is the most suitable choice to be assigned as the origin since its position remains the same with respect to the moving blades. Using one of the blades as an axis, any point of interest on the OTW blades can be localized accurately [107]. However, this approach is only suitable to be carried out from a distance, purposely for visual inspection. For IMR with close contact operations, as soon as the origin is located, the flight path towards any point of interest can be planned. However, without the GNSS fused with the IMU navigation system, the vision-guided UAV will lose its positioning while approaching a large structure, as the view will be zoomed in. To overcome this matter, the visual-inertial odometry technique using point and line features can be implemented [84]. This approach has enabled a UAV to work robustly in strong electromagnetic interference, low-textured and illumination-change windfarm.

Since the visual-inertial odometry technique is based on dead-reckoning localization, it is still susceptible to drift from the true localization due to the accumulation of errors in relative motion estimation [111]. Using a collaborative approach may solve this problem by constantly updating the position of the UAV with respect to the WT origin. This can be implemented by deploying another UAV or a swarm of UAVs that has the capability of monitoring the UAV working near the OWT with full visualization [112]. The monitoring UAVs, having a direct view of the OWT origin with respect to the working UAV, might transmit the correct position through radio frequency communication [113,114], if possible, or using a visible light communication [115], which is immune to interference from electromagnetic sources. Another possible method is by using ultraviolet direction and

ranging [116] to steer another UAV in a three-dimensional position and orientation relative to the monitoring UAV.

Nevertheless, prior knowledge of a WT structure, for example, the blade length or the tower height, is needed to determine the absolute localization of the WT, which is one of the drawbacks when using visual-based measurement. Nonetheless, with recent advancements in the twin digital technology of wind farms [117], prior knowledge on the OWT structures can easily be retrieved. On the other hand, the visual SLAM methods do not require prior knowledge of the WT structures. Using a specific algorithm with a generated synthetic depth map to localize a UAV within an environment [85], a WTB can be detected at its centre (hub). However, since the UAV uses a flight plan guided by the IMU to approach the WTB, it cannot prevent drifting errors from occurring. Again, this can be solved using the collaborative approach for visual-based navigation as previously discussed.

Another method, a LiDAR-based SLAM approach, has been proposed for WTB localization [108]. Within this approach, the proposed algorithm manages to map a WTB with greater performance when the proximity is around 0.5 m distance to the blade. Localization is achieved by marking the origin to the centre of the scanned blade. However, the centre of the origin can be set to the hub as the WT centre when full scanning is completed. For OWT blade inspection, the LiDAR-based SLAM can be initiated by initially guiding a UAV close to a WTB, followed by efficient LiDAR scanning [118]. With this approach, post-processing of the WTB inspection data can be stored, where analysis of the recorded inspection data can later be made and future revisiting for maintenance and repair operations is also made possible. For future revisiting, a LiDAR-based SLAM UAV should be initially guided to the recorded origin location (hub) where a recorded map from the previous visit can be restored and matched using a specific algorithm [119]. The UAV can then approach any point of interest on the map, starting from the origin point using the improved Reference LiDAR-based Odometry and Mapping (R-LOAM) approach [120].

Using the LiDAR-based SLAM technique, 1 to 2 centimetre-level mapping accuracy can be achieved [121]; hence, future revisiting becomes easier. However, this type of localization requires high computational loads [122] which can easily drain the UAV energy [123], thus causing shorter flight time for IMR operations. It can also add extra weight to the UAV [124], thus reducing the payload capacity for carrying IMR tools. Nonetheless, using collaborative strategies as discussed in the previous section, the UAV can make use of the USV as a nearby wireless data processing and storage system provider [60,61], as well as a power hub to prolong the UAV operation time [50,80].

In addition, the LiDAR-based SLAM technique, which is not affected by magnetic interference, can be operated while a WT is fully operational. However, most of the time, a WT shutdown is necessary during maintenance and repair operations. Due to this fact, a UWB positioning system might be preferred as it can perform better under the temporary shutdown of an OWT during IMR works. Using a real-time accurate solution, a localization system on an OWT can easily be associated with the UAV localization system. Moreover, future revisiting is also more straightforward than the LiDAR-based localization since the collaborative UWB localization technique (integrated with GNSS-PPP and local WT IMUs) can point directly to the point of interest within the WT absolute coordinate system, instead of needing to find the local origin in the initial step.

Apart from addressing the need for high-accuracy localization and navigation with respect to IMR activities close to a WT, it is also needed for other collaborative purposes such as for landing, docking, or recovery operations. Since UAVs have limited energy capacity due to current battery technology, their batteries need to be recharged frequently to prolong their operating times at sea. Since there is no charging station at sea, implementing a USV as a mobile charging station is seen as the current most practical solution. However, landing and docking a UAV onto a USV is more challenging at sea compared to onshore due to the presence of waves, sea current and higher wind speed.

For a UAV to land on a USV, the most critical part is the ability to detect the location of the landing platform on a USV. To solve this matter, a visual-based method has been

proposed to detect and identify a properly designed fiducial marker placed on a USV [44]. This fiducial marker enables UAV landings in the absence of the global positioning system by providing relative pose estimates between UAVs and the landing platform. Using a specific 2D pattern, a UAV will be able to view and determine its own relative position to land on a USV from a reasonable distance of 3.74 m. Further improvement to this method [45] has been made by proposing a three-stage fiducial marker that provides more visual exposure while landing with the first detection at 4.6 m and further detection at 1.4 m. Subsequently, both methods adopted further control strategies to finally land a UAV using a PID controller for accurate landing.

However, the visual-based platform detection method is ineffective under adverse weather conditions such as rain or fog due to poor visibility. Hence, an alternative method using an infrared beacon and an infrared detector is proposed for the UAV landing guide [49]. In this method, an infrared detector is placed on the bottom of a UAV whilst the infrared beacon is placed at the centre of the USV landing platform. Under this configuration, the UAV can localize the infrared beacon for precise landing. Nevertheless, offsets in the landing position can be observed from the experimental outcome due to the wind effect during the landing tests.

Due to the dynamic sea environment, especially from high wind and sea waves, robust control strategies utilizing sensors and actuators have been proposed to ensure the safe landing and transportation of a UAV on a USV [48]. Using hierarchical landing guide points generated using four ultrasonic sensors on a USV, a UAV can be navigated safely towards the last guiding point on the USV's platform where a multi-ultrasonic joint dynamic positioning algorithm is activated to ensure accurate landing. For secure transportation, the landing platform can automatically fold up to lock and carry a UAV.

While a UAV landing directly onto a USV is seen as the most practical solution for recovering a UAV, a different approach using a robotic recovery system has been proposed [71]. Instead of guiding a UAV precisely to land on a USV platform, this method allows a UAV to land smoothly on a nearby water surface without further computational burden to position itself precisely on a USV platform. The USV will then locate the position of the UAV using RTK GPS and computer vision, followed by recovering the UAV by deploying a submersible perforated net. The UAV will then be lifted and locked onto the USV platform. Apart from planned recovery, this method is extremely useful in the case of emergency recovery of a fallen UAV due to unexpected malfunctions during operations. However, it should be noted that this approach requires a UAV to keep afloat during recovery and it should basically be waterproof and saltwater-resistant.

### 3.3. Collaborative Strategies to Prolong the Operation Time

Extending a UV operating time can be considered another important factor to be considered for offshore IMR activities. Currently, UAVs or UUVs depend heavily on a crewed vessel for energy supply, either by using tethered power cables or onboard chargers for the batteries. The most conventional method to achieve an uninterrupted power supply is by using a tethered system from a USV through a power umbilical cable. Although tethered systems are often associated with issues such as cable entanglement or jamming, significant strategies to optimize cable length and tension may however prevent these problems [53]. For OWT operations, using tethered UAV or UUV powered by a USV may solve the energy supply issue, especially when operating power-draining tools for maintenance or repair operations. While using a tethered system might solve the energy issue, however, there are many other weighing factors that might hinder its use, such as the manoeuvring flexibility around the OWT structure and the limitation of the cable's length in covering the whole WT structure. Thus, a wireless battery-powered UAV or UUV is often seen as a more practical solution to operating near a complex OWT structure.

There are many strategies being proposed and developed for automatic UAV battery charging, beginning with a battery swapping system [125,126] and during flight battery charging [127] to wireless charging [52,128,129]. In a battery swapping system, autonomous

hot battery swapping has been used to ensure no data losses during the process [125]. In this process, a UAV needs to land precisely on a ground charging station consisting of a battery carousel platform. As soon as the landing takes place, direct electrical contacts between the charging platforms will supply enough energy to the UAV while a battery swapping takes place. A grasping mechanism will pull off the depleted battery and place it into an empty slot on the carousel. The carousel will then rotate to align a fresh battery to be inserted into the UAV battery slot.

Another battery swapping method uses an inverted docking station as a battery swapping platform [126]. In this method, a quadrotor UAV will precisely attach itself to an inverted platform on a ceiling where further battery swapping using a gripper will take place. The purpose of introducing the inverted docking station is to allow a UAV to dock safely with a load attached below it. This method could be useful in the case where a load or a tool with different sizes would not enable a UAV to dock on a charging platform. However, for such a system to be implemented on a USV would require a special platform or space with a ceiling to be constructed.

Another advanced method to ensure continuous UAV operation is by using a battery switching method during flight [127]. In this system, a quadcopter UAV (primary UAV) with a docking platform attached on top of it will be approached by a small UAV (secondary UAV) carrying an extra battery. The small UAV equipped with four docking legs will then mate with the primary UAV while still in the air. While docking, the mating legs function as connectors to power the primary UAV. At this stage, through a carefully designed battery switching circuit, the primary UAV will switch off its own battery and totally rely on the secondary battery. As soon as the second battery is fully depleted, the main UAV will switch back to its own battery and the small UAV will return to its base. This battery switching process is continuously repeated, with many small UAVs successively docking and being retrieved to provide extra energy supply to the main UAV. Eventually, the main battery will be fully depleted. To achieve a longer operating time, the interval time between successive changing of the secondary UAVs needs to be minimized as little as possible.

While the battery swapping and battery switching system might be useful, wireless charging of a UAV has gained significant attention due to the flexibility it brings to the charging process. Using a capacitive power transfer technology [128], a UAV only needs to land on a designated platform, without the need for precise orientation or alignment. This could significantly reduce complex control strategies and mechanisms just for the purpose of battery alignment to initiate the charging process. Furthermore, a wide charging area is made possible with this wireless charging technology. For a UAV docking on a USV [52], this technology can further ease the process, as swaying effects on a USV make docking strategy even harder when compared to docking on land. The wireless charging method for UAVs is further applied to prolong the UAV flight time by placing multiple landing pads along a flight path [129]. This strategy is useful for UAVs operating for long-distance surveying or in wide area coverage.

Almost all automatic charging strategies can be implemented by USV to power up a UAV; however, the most practical solution would be the wireless charging method due to its simplified mechanical arrangement which does not require a UAV to properly align or be fitted to a charger. The wireless charging method uses inductive coupling to enable flexible contact between a UAV and a charging platform. Due to this practicality, a USV can easily charge a docked UAV on its charging platform without the complex control of automated mechanical parts.

### 3.4. Collaborative Communication Strategies

Although a commercial unmanned solution for offshore operations already exists [63], however, a communication issue is believed to be the main factor that prevents it from being implemented for most OWT farms. With a maximum range of just 5 km, a communication system based on a Wi-Fi connection, although superior for real-time control and data transmission, cannot be used for most offshore wind farms due to non-existent coverage.

For OWT inspection, currently crewed vessels are still being used to transport UAVs near offshore assets [130]. To switch to full unmanned operations, a USV will normally be deployed to carry UAVs without any human operator. Within this arrangement, human operators can be kept in a loop at the GCS, either to monitor a fully autonomous unmanned system or to remotely control the UVs. Using cellular 4G or 5G networks, communication between the GCS with all UVs can be established. However, as wind farms have rapidly been located even further to sea, the probability of zero coverage for 4G or 5G networks is high. UAVs operating within the IMR operations will rely heavily on a reliable and uninterrupted data transmission system. Without sound communication infrastructures, the transmission of IMR data to a GCS will be problematic. Unless proper structure for a cellular network is made available for offshore wind farms, the IMR operations will have to rely on the next best communication solution, which is through satellite networks.

Globally, the average distance to shore for offshore wind farms is 18.8 km [131]. Hence, most wind farms rely on a satellite communication system for monitoring purposes. However, a normal satellite communication system has a significant transmission lag in the order of a few minutes to hours due to the high latency issue [132]. Due to this limitation, it is impossible to control a UAV remotely from the GCS, as real-time communication is not available. However, recent advancement in BVLOS satellite communication which utilizes Inmarsat SwiftBroadband satellite service has enabled real-time navigation and control with live video transmission [133]. With a data transmission speed of 432 Kbps and weighing just 1.45 kg in a compact $24 \times 16 \times 6$ cm$^3$ enclosing, it is a desirable embedded solution for offshore drones. However, like any other satellite-based communication system, operating near a tall WT structure may affect data transmission due to the loss of signals. To overcome this issue, a collaborative approach between a UAV and a USV can be devised. A USV may be stationed near an OWT while taking into consideration a safe distance for receiving optimized satellite signals.

To establish a collaborative communication system, a UAV and a USV equipped with a satellite communication device can be linked with a GCS through a distributed dynamic network topology within an ad hoc network [46]. Utilizing this network, data from the UAV can initially be transmitted to the USV using wireless communication modules (2.4 GHz Wi-Fi module), which are then further transmitted from the USV to the GCS through satellite communication. Vice versa, commands to control the operation of the UAV can be transmitted from the GCS through satellite communication via the USV.

The distributed dynamic network can be further expanded for underwater communication involving the UUVs. Similar to the UAV, the UUV may also be controlled from a GCS through the satellite communication system via the USV. However, instead of using wireless radio frequency for data transmission, UUVs mostly rely on an acoustic underwater communication system. Unfortunately, data transmission using acoustic transmission for the purpose of IMR operations cannot be performed in real time due to the slow transmission rate of the acoustic signals. Nevertheless, this can be achieved via short-range wireless transmission using underwater radio frequency communication or optical communication at data transmission rates ranging from a few Mbps to Gbps [134]. A UUV as an inspection-class ROV can be piloted wirelessly from a control station up to a range of a few tens of metres and can perform semi-autonomous operation up to a range of a few hundreds of metres. However, without direct data transmission between the control station and the UUV as a result of transmitting data via the satellite communication system through the USV, noticeable delays and poor quality of data transmission can be expected.

As an alternative for the USV-UUV collaboration, a UUV may also collaborate directly with a UAV for various reasons such as to achieve high-speed mobility in tracking and updating a UUV [59] or for guiding a UUV directly from above [58]. There are very few applications of direct collaboration between a UAV and a UUV due to the communication barrier, as radio frequencies are heavily attenuated through a water medium. Usually in such a collaboration, a UUV is the one that requires cooperation from a UAV to determine its own localization with respect to the GNSS position. For such a collaboration, normally a

water-air communication relay system is needed which can be installed on a vessel, buoy, or even on a USV. Previously, a UAV could only communicate with a UUV through a control station [58]. However, with recent advancements in UAV technology, UUVs are now able to communicate directly with a UAV that is equipped with sonar for underwater communication [59]. These UAVs are capable of landing smoothly and floating on a water surface to establish water-air communication.

Nonetheless, the benefits of deploying a USV as the intermediary between a UUV and a GCS are greater when compared to direct UAV-UUV collaboration. In a USV-UUV collaboration, the USV not only solves the communication issue for the UUV but can also be used as a recovery and docking station, guidance for underwater localization and navigation, an energy provider, or a charging station. To ease the computational burden while processing large amounts of inspection data, the USV can also be deployed as a data processing and storage hub.

## 4. Redundancy for Collaborative UVs

Within the OWT IMR operations, the reliability and availability of unmanned systems are very important to minimize the risks associated with safety, poor performance, and data security. While a collaborative system may improve the operation of current unmanned systems, it may also replicate several failure modes that exist in a single operating UV. In the worst-case scenario, it may amplify the degree of severity of the failure modes which could result in more severe damage to the vehicles or assets, loss of operating time, or critical data breach. To prevent or reduce the risks of these failure modes, a redundant system for collaborative UVs should be introduced.

### 4.1. Potential Failure Modes of a Collaborative System

The common failure modes for a single UV may be caused by a component failure (sensors, actuators, software, and hardware), human error (poor navigation, faulty setup, and poor maintenance), or environmental factors (bad weather, unsuitable ambient temperature) [130]. Other common failure modes may also be caused by positioning errors due to GNSS issues [135], electromagnetic interference [136], or security breaches from spoofing activity [137].

Within a collaborative system, these failure modes may be amplified due to the involvement of multiple UVs instead of just a single UV. When UVs are working together in close proximity, there is greater potential for collision between them that might be caused by the common failure modes. For collaborative UVs performing a mission or an IMR operation, a single UV malfunction may halt the entire operation, and eventually will cause a significant loss of valuable operating time. Poor coordination or coupling between UVs might also cause losses or damage to one or more of the UVs, for instance during launching, docking, or transportation of a UAV by a USV.

### 4.2. Redundancy Strategies

In designing a redundant system, Failure Modes and Effects Analysis (FMEA) is initially carried out to identify the root cause of the potential failures [130]. Strategies of sensor fusion using two GNSS receivers with two IMUs have proven to be reliable in providing robust positioning for a UAV [138]. Other sensor fusion methods that involve aiding a GNSS with visual-inertial odometry [135] and fusion with IMU and LiDAR [139] have also been reliable in providing stable localization and navigation of a UAV.

Regarding collaborative UVs within the IMR OWT operations, since they are still in their infancy stage, the FMEA, to the best of the authors' knowledge, has not been carried out yet. However, through modelling and simulation studies of collaborative UVs for the IMR OWT operations, the FMEA as a potential resource for introducing redundancy can be carried out with proper validations.

From a different point of view, strategic planning and deployment of multiple UVs may as well establish a redundant system within the collaborative system. As previously

discussed in Section 3.2, a USV may provide additional GNSS guidance from a safe distance to increase the reliability of the UAV navigation. Hence, collisions can be prevented by addressing the issue of poor localization and navigation system. In the case of a malfunctioning UAV component, a possible redundancy strategy might be introduced by using a component replacement station which is placed on a nearby USV. However, a sophisticated mechanism would be required for this method. Furthermore, as discussed in Section 3.4, a collaborative communication system through a private ad hoc network may also reduce exposure to cyber-attack that usually occurs through penetration through an external gateway. Other redundant strategies through collaborative efforts can be further devised to address the failure mode associated with the implementation of collaborative UVs.

## 5. Conclusions

Collaborative UVs approaches have the potential to be the enabler for implementing unmanned IMR operations within the OWT industries. Hence, this paper reviewed the collaborative strategies that can be implemented using several types of UVs for the OWT IMR operations. This review also highlighted the potential solutions and improvements a collaborative UVs approach may bring. To conclude, this review shows that:

- Unmanned vehicles such as UAVs, UUVs, or USVs have a very limited capability to perform unmanned IMR operations for OWT when on their own, with the exception of carrying out visual inspections and simple NDT inspections.
- A swarm of drones or homogeneous UVs may accomplish complex IMR tasks but they still depend on USV as a power charging station, to change tools, as a communication hub, etc.
- Many collaborative strategies integrating the UAVs, USVs, UUVs, and crawler robots can be implemented as presented in Table 3, which depends on the degree of task complexity.
- USVs as sources of accurate global positioning (GNSS) may provide reliable and better localization systems for UAVs and UUVs to improve their existing localization system.
- UWB collaborative localization system is currently the best approach for UAVs to work close to an OWT, providing the OWT is shut down temporarily. However, the LiDAR-based SLAM technique has the potential to be implemented while the WT is fully operational.
- The implementation of a USV within a collaborative unmanned network has the potential to solve a number of issues surrounding the unmanned IMR OWT operations, including performing complex tasks, achieving better localization and navigation, prolonging operation time, and establishing better communication.
- Satellite communication system is currently the best communication platform for OWT sectors due to its geolocation. However, the introduction of 5G networks at offshore wind farms is slowly opening up the opportunity for a better communication system.
- At present, there is no redundancy strategy available for the collaborative UV; however, the redundant systems that are currently being deployed for a single UV can be further adapted for the collaborative unmanned systems. Collaborative approaches may also be used to address failure modes in designing redundant systems.

*Suggested Future Research Directions*

The authors would recommend the following future works:

- A collaborative LiDAR-based SLAM localization algorithm can be developed to enable zero-downtime IMR operation for the OWT. This approach can minimize losses of OWT operation due to unwanted shutdown time.
- UVs in terms of a crawler robot or a UAV that can stick firmly to a WTB while in operation without suffering positioning error or odometry failure should be further explored.
- Investigation of a reliable localization and navigation system for UAV utilizing mobile UWB anchors and fixed IMU on an OWT for swaying floating OWT can be further explored.

- Investigation into human-UVs collaboration using tele-operated manipulators for high-skilled OWT maintenance and repair tasks from an inland control station should be carried out.
- Redundancies to address potential failure modes for the collaborative UVs should be further explored and such systems should be devised to ensure the reliability and availability of collaborative UVs for the IMR OWT operations.

**Author Contributions:** Conceptualization, M.H.N. and S.S.; investigation, M.H.N.; writing—original draft preparation, M.H.N.; writing—review and editing, S.R., M.G., A.K. and R.S.; visualization, M.H.N., S.S., A.K., S.R., M.G. and R.S.; supervision, S.S. All authors have read and agreed to the published version of the manuscript.

**Funding:** This research was carried out as part of the Cornwall FLOW Accelerator (Project no. 05R19P03188) and Marine-i 2 (Project no. 05R18P02816) projects, which are partly funded by the European Regional Development Fund (ERDF) as part of the European Structural and Investment Funds (ESIF) Growth Programme 2014-20.

**Institutional Review Board Statement:** Not applicable.

**Informed Consent Statement:** Not applicable.

**Data Availability Statement:** Not applicable.

**Conflicts of Interest:** The authors declare no conflict of interest.

## Abbreviations

The following abbreviations are used in this manuscript:

| | |
|---|---|
| WT | Wind Turbine |
| WTB | Wind Turbine Blade |
| OWT | Offshore Wind Turbine |
| UV | Unmanned Vehicle |
| UAV | Unmanned Air Vehicle |
| UUV | Unmanned Underwater Vehicle |
| USV | Unmanned Surface Vehicle |
| ROV | Remotely Operated Vehicle |
| BVLOS | Beyond Visual Line of Sight |
| GCS | Ground Control Station |
| AUV | Autonomous Underwater Vehicle |
| NDT | Non-Destructive Testing |
| IMR | Inspection, Maintenance, and Repair |
| O&M | Operation and Maintenance |
| UWB | Ultra-wide Band |
| GNSS | Global Navigation Satellite System |
| SLAM | Simultaneous Localization and Mapping |
| LiDAR | Light Detection and Ranging |
| IMU | Inertial Measurement Unit |
| RTK | Real-time Kinematics |
| PPP | Precise Point Positioning |
| PPK | Post-processed Kinematics |
| R-LOAM | Reference LiDAR-based Odometry and Mapping |
| FMEA | Failure Modes and Effects Analysis |

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
