# Peer review of "Collaborative Unmanned Vehicles for Inspection, Maintenance, and Repairs of Offshore Wind Turbines"

_drones, doi:10.3390/drones6060137_

Round 1

Reviewer 1 Report

The authors revised the article according to the given comments by me. Now I recommend to publish it in its present form.

Author Response

We would like to express our gratitude and are very thankful for your time and valuable insights in improving our manuscript.

Reviewer 2 Report

I couldn't see enough modifications from the last submission and all my comments are still unanswered.

Author Response

From the previous submission, the following modifications have been made to address the comments:

Basically, this review only intends to highlight the contributions of the existing works that are directly involved with proposing or developing collaborative approaches for unmanned vehicles. It is from the authors' initial understanding that most cited articles that have not been discussed are only cited to mention the existence of the works that have been done that can lead to the deployment of the collaborative unmanned vehicles. To address the comment from the referee, we further discussed the contribution of each paper and how they differ from each other which is summarised in tables as an effective way of presenting the contributions.

  1. Moving forward, significant efforts which are highly viewed as the most viable solution are currently being developed, which relate to the integration of multiple types of Uvs to collaboratively perform unmanned OWT IMR (e.g, UAV and USV [44-49], USV and UUV [50-53], UAV and UUV [54-55], or a combination of all Uvs [56-59]).”

Table 1 has been presented to highlight the contributions of each paper in terms of the different methods that have been carried out to achieve the collaborative purposes.

  1. “A tool-changing UAV might as well be a UAV that is equipped with a manipulator [63-66], an arm [67-68], or multiple arms [37, 43, 69] for more dexterous manipulation during maintenance or repair works.”

Table 2 has been presented to highlight the contributions of each paper in terms of the unique feature proposed by each paper and the potential UV collaboration they can bring to the implementation of IMR operations within the OWT sector.

For this submission, we further identified cited articles that are worth to be discussed further as previous efforts are insufficient. Literature studies that have been discussed further are listed below:

  1. Further discussion on how careful sequence of operation is carried out by [76].
  2. Further discussion on the mechanism of two swivel arms as proposed by [77].
  3. Further discussion on how a crawler robot can be deployed on a WTB [24].
  4. Table 4 is introduced to summarize the contribution of several visual localization methods working near a wind turbine.[83-87].

Reviewer 3 Report

The revised manuscript has addressed most of my earlier concerns. However, I do not consider the "limited time and knowledge of the author" to be a suitable excuse for not addressing issues of redundancies. If a system fails and has no redundant system that can be used instead (even with reduced functionality), then you have an accident (or at least are likely to have one). In particular, GNSS failure is particularly common. If the author really does not want to address this in the body of the manuscript, then there should be a limitations section that highlights the importance of exploring redundancies within any collaborative system such that there are ways of managing relatively common failure modes. You can then say this wasn't done in the manuscript because you were focussed on concepts of operation (or some other similar argument). I do not consider that the manuscript should be published if it does not consider failure modes and the need to have thought about redundancies with any collaborative approach. Otherwise the authors would be advocating for systems that could pose risks to airspace users, equipment, and any humans in the operating vicinity.

Once the issue of redundancies has been addressed then this manuscript should be ready for publication.

Author Response

Based on the reviewer's motivation and further guidance, Section 4 has been added to briefly address the potential failure modes that may exist for the collaborative UVs in the IMR OWT operations. Although very limited, the authors have suggested a few potential redundancy strategies to address the failure modes. The authors also highlighted the importance of redundancies that should be further explored in the section for future research directions. These changes to the manuscript have been reflected as well in the abstract, introduction and conclusion sections.    

Round 2

Reviewer 2 Report

I'm sorry but I think contributions are not enough and the novelties are not considerable.

Author Response

Regarding the contributions, the authors would like to emphasise that the review has been made to highlight that currently there is no collaborative unmanned vehicles being used for the inspection, maintenance and repair (IMR) within the offshore wind turbine sector. It is understood that switching to using unmanned vehicles could reduce carbon emissions significantly and replacing humans in hazardous environments. 

In terms of novelties, no other papers have actually reviewed the current implementation of collaborative unmanned vehicles within the OWT sector, instead just the homogenous use of drones for wind turbine blade inspection.

Furthermore, this paper further proposes possible collaborative strategies that can be implemented to booth the unmanned IMR operations for the OWT using collaborative unmanned vehicles.

Therefore, it is hoped that the referee could consider again the decision. Thank you for your time and kind consideration.

This manuscript is a resubmission of an earlier submission. The following is a list of the peer review reports and author responses from that submission.

Round 1

Reviewer 1 Report

The artile having titled as "Collaborative Unmanned Vehicles for Inspection, Maintenance, and Repairs of Offshore Wind Turbines", relates to my area of interest. Moreover, I suggest the following changes in the article to the authors which are as follows; 

a. Abstract is little bit weak need to add the main motivation line in it

b. In introduction whats the main contibutions as well as arrangement of the manuscript is mising.

c. Diagramatic presentation is required in this article the figures are not to the point.

d. What is the problem statement and proposed solution

e. Authors must add some data and comparitive analysis in this article in order to achieve the readibility of this article.   

My recommendations are rejet and resubmit with major changes

Reviewer 2 Report

In this review manuscript, the collaborative strategies for using unmanned vehicles for inspection, maintenance, and repair operations on offshore wind turbines are addressed. In my view, the authors' most important contribution is Table 1, which provides a classification and comparison of collaborative strategies. However, most of the manuscript consists of citing papers without in-depth discussions, which is expected of a review article. for example:

"Moving forward, significant efforts which are highly viewed as the most viable solution are currently being developed, which relate to the integration of multiple types of UVs to collaboratively perform unmanned OWT IMR (e.g, UAV and USV [44-49], USV and UUV [50-53], UAV and UUV [54-55], or a combination of all UVs [56-59])."
or
"A tool-changing UAV might as well be a UAV that is equipped with a manipulator [63-66], an arm [67-68], or multiple arms [37, 43, 69] for more dexterous manipulation during maintenance or repair works."

About 10 references are cited in these examples without indicating their main contributions or comparisons between them. In fact, most of the cited articles were not discussed or cited and were only mentioned with the information that could be understood even from the title of the article (without reading the entire body of the article).

Reviewer 3 Report

This paper was a fascinating read and has been well put together. It is suitable for this journal because of its practical contribution, which makes for a good review article. I noticed a few spelling/grammar issues and had a couple areas where I felt more clarity could be provided. However, with only minor tweaks this paper will be ready for publication.

  1. Line 90 - seal level should be sea level
  2. Line 108 - given that a numbered system is used, citing a course by saying "developed by [37]" doesn't read well. You can delete by and keep the [37] and it will still make sense.
  3.  Line 160 - should be UAVs rather than UAV.
  4. Lines 187-188 "might as well be" has a different connotation to what the authors are trying to convey. I suggest "may also be".
  5. Lines 199-200, same concern regarding how [76] has been cited. If you remove "as designed in" and move [76] to the end of the sentence then it will read better
  6. Lines 223-225, I suggest you reword this to something like "Table 1 presents a summary of the benefits and challenges associated with different collaboration strategies." The current wording is a bit confusing. 
  7. Table 1 - in the challenges column, "required" and "require" are used. Because the strategy is singular, it should be "requires" for all instances.
  8. Table 1 - third row of USV-UAV strategy, in challenges column, "Require uninterrupted communication with GC station through USV. " should be "Requires uninterrupted communication with the GC station through the USV".
  9. Table 1, first USV-UAVs row, in benefits column, "Can perform multi-tasks operations in sequence using a different set of tools on multiple UAVs." should be "Can perform multiple tasks in sequence using different sets of tools across several UAVs."
  10. Table 1, final row of USV-UAVs strategy, in the challenges column, "difficult" needs to be capitalised. 
  11. Table 1, final row of USV-UUV strategy, in the challenges column, "Require uninterrupted communication with GC station through USV." should be "Requires uninterrupted communication with the GC station through the USV." 
  12. Table 1, first row of USV-UUVs strategy, in the challenges column, "Require a UUVs recovery and storing mechanism." should be "Requires a recovery and storing mechanism for the UUVs." AND
  13. Table 1, multiple rows of the USV-UUVs strategy under the challenges column, "Tethered UUV might be tangled with each other" should be "Tethered UUVs might become tangled with one another."
  14. Table 1, second row of USV-UUVs strategy, under the challenges column, "Require a UVMSs recovery and storing mechanism." should be "Requires a recovery and storying mechanism for the UVMSs"
  15. Table 1, next row below in challenges column, "Require a universal UUVs recovery and storing mechanism." should be "Requires a universal recovery and storing mechanisms for the UUVs".
  16. Line 253, same issue regarding citing sources. Rather than me restating this issue, I suggest the authors re-read their manuscript to identify instances where the use of the numbered referencing system makes the sentences sound awkward. Best practice is to just say what you a citing and then put the numbered reference at the end of the sentence. You can usually write how the authors have done with other styles (e.g., APA or Harvard", but is doesn't really work with numbered systems.
  17. As far as I understand, UWB also has a limited range. It is a really good suggestion, but it may be worth highlighting the range to confirm that the altitude the UAVs would be flying at would be within range.
  18. As a general comment, I think that the paper needs to address redundancies in the event of failure of different navigation systems. It is not uncommon that systems do fail, and so having more than one built in makes sense. For example, from operational experience I have seen cases with RTK has failed - if its dual RTK, then you can continue with the same approach, otherwise software may be used to work out a heading and return to home (i.e., by flying some pattern to work out where it is based upon previous information). Failure modes and redundancies would make a useful contribution to the manuscript.
  19. I was also curious why for UAVs there is no mention of positioning using infrared cameras and beacons positions on the USVs. This is probably an inferior approach to UWB, but may be worth mentioning?

Aside from the above points (that mostly relate to grammar), the manuscript is well-written and will make a useful contribution once published. Well done to the authors for an excellent submission.